# Pluronic-F-127-Passivated SnO_2_ Nanoparticles Derived by Using *Polygonum cuspidatum* Root Extract: Synthesis, Characterization, and Anticancer Properties

**DOI:** 10.3390/plants12091760

**Published:** 2023-04-25

**Authors:** Badr Alzahrani, Abozer Y. Elderdery, Nasser A. N. Alzerwi, Abdullah Alsrhani, Afnan Alsultan, Musaed Rayzah, Bandar Idrees, Fares Rayzah, Yaser Baksh, Ahmed M. Alzahrani, Suresh K. Subbiah, Pooi Ling Mok

**Affiliations:** 1Department of Clinical Laboratory Sciences, College of Applied Medical Sciences, Jouf University, Sakaka 72388, Saudi Arabia; 2Department of Surgery, College of Medicine, Majmaah University, P.O. Box 66, Al-Majmaah 11952, Saudi Arabia; 3Department of Surgery, King Saud Medical City, Riyadh 12746, Saudi Arabia; 4Department of Surgery, Prince Sultan Military Medical City, P.O. Box 7897, Riyadh 11159, Saudi Arabia; 5Aseer Central Hospital, Abha 62523, Saudi Arabia; 6Iman General Hospital, Riyadh 12684, Saudi Arabia; 7Centre for Materials Engineering and Regenerative Medicine, Bharath Institute of Higher Education and Research, Chennai 600073, India; 8Department of Biomedical Science, Faculty of Medicine & Health Sciences, Universiti Putra Malaysia, Serdang 43400, Selangor, Malaysia

**Keywords:** pluronic F-127, tin oxide nanoparticles, anticancer, *Polygonum cuspidatum* root extract, HepG2 cancer cells

## Abstract

Nanotechnology has emerged as the most popular research topic with revolutionary applications across all scientific disciplines. Tin oxide (SnO_2_) has been gaining considerable attention lately owing to its intriguing features, which can be enhanced by its synthesis in the nanoscale range. The establishment of a cost-efficient and ecologically friendly procedure for its production is the result of growing concerns about human well-being. The novelty and significance of this study lie in the fact that the synthesized SnO_2_ nanoparticles have been tailored to have specific properties, such as size and morphology. These properties are crucial for their applications. Moreover, this study provides insights into the synthesis process of SnO_2_ nanoparticles, which can be useful for developing efficient and cost-effective methods for large-scale production. In the current study, green Pluronic-coated SnO_2_ nanoparticles (NPs) utilizing the root extracts of *Polygonum cuspidatum* have been formulated and characterized by several methods such as UV–visible, Fourier transform infrared spectroscopy (FTIR), energy dispersive X-ray (EDAX), transmission electron microscope (TEM), field emission-scanning electron microscope (FE-SEM), X-ray diffraction (XRD), photoluminescence (PL), and dynamic light scattering (DLS) studies. The crystallite size of SnO_2_ NPs was estimated to be 45 nm, and a tetragonal rutile-type crystalline structure was observed. FESEM analysis validated the NPs’ spherical structure. The cytotoxic potential of the NPs against HepG2 cells was assessed using the in vitro MTT assay. The apoptotic efficiency of the NPs was evaluated using a dual-staining approach. The NPs revealed substantial cytotoxic effects against HepG2 cells but failed to exhibit cytotoxicity in different liver cell lines. Furthermore, dual staining and flow cytometry studies revealed higher apoptosis in NP-treated HepG2 cells. Nanoparticle treatment also inhibited the cell cycle at G0/G1 stage. It increased oxidative stress and promoted apoptosis by encouraging pro-apoptotic protein expression in HepG2 cells. NP treatment effectively blocked the PI3K/Akt/mTOR axis in HepG2 cells. Thus, green Pluronic-F-127-coated SnO_2_ NPs exhibits enormous efficiency to be utilized as an talented anticancer agent.

## 1. Introduction

Nanotechnology is a rapidly intensifying discipline of study that focuses on synthesizing, characterizing, modifying, and utilizing nanomaterials for their enormous potential across multiple industrial sectors [1,2]. As opposed to their bulk counterparts, the exceptionally high surface-area-to-volume ratio and nanometre dimension of these materials may lead to entirely novel or increased electrical, magnetic, optical, catalytic, and antibacterial functionalities [3]. SnO_2_ is one such nanomaterial that has attracted the most interest among researchers because it has an extensive scope of applications that includes optoelectronic materials, dye-based solar cells, gas and light sensors, lithium-ion battery electrodes, LEDs, catalysis, medical devices, and antistatic coatings [4]. SnO_2_ NPs from the *Annona squamosa* dried peel aqueous extract was evaluated for their cytotoxicity against the liver cancer HepG2 cells [5]. In this work, the first attempt was made to formulate the plant-derived Pluronic-F-127-encapsulated SnO_2_ nanoparticles using *Polygonum cuspidatum* root extract was performed to assess their effectiveness for inhibition of anticancer in in vitro studies.

Researchers across the globe have been exploring a variety of techniques for synthesizing nanoparticles (NPs) because of their distinctive physicochemical characteristics and potential applicability. Chemicals that are potentially harmful to the environment and public health are employed in both the chemical and physical techniques employed to fabricate metal and metal oxide NPs. These techniques are costly and complicated [6]. Instead, the biological approach offers higher reliability, eco-friendliness, cost-effectiveness, high yield, and easy processing without negative environmental effects, and thus, interest in its application has significantly expanded over the past decade. Moreover, during synthesis, active substances found in plant resources also serve as reducing, capping, and stabilizing agents [7].

While developing nanodrugs for therapeutic delivery, the development of stable systems for efficient drug delivery is crucial. As a result, efforts to develop delivery techniques have turned toward more stable and controlled systems, including polymeric nanoparticles. Polymer–protein conjugates, polymer–small molecule conjugates, dendrimers, polymeric vesicles, polymeric nanospheres, and polymeric micelles are only a few examples of available polymeric nanomedicine systems for drug delivery [8]. The core material of these polymer-based NPs can be selected to enhance drug encapsulation efficacy. Specifically, the focus on polymeric amphiphiles as core constituents has grown lately as these polymers could be employed with both hydrophobic and hydrophilic molecules. The amphiphilic block copolymer, Pluronic, has received significant attention in the drug delivery sector [9].

Pluronic F-127 (PF-127), also called as poloxamer-407, is a triblock amphiphilic copolymer, which comprises central hydrophilic PEO (poly-ethylene oxide) chains flanked by two hydrophobic PPO (poly-propylene oxide) chains [10]. Owing to its high solubilizing ability, biocompatibility, reverse gelation, and minimal toxicity, PF-127 is regarded as an appropriate medium for drug delivery through several parenteral and non-parenteral routes [11,12]. The functional characteristics of these polymers have been extensively studied in the last decade. This has result in the formation of a array of systems with broad use as drug delivery and release vehicles for proteins, nucleic acids, and peptides. Topical administration of lidocaine, topical pain-relieving/anti-inflammatory drugs, anticancer drugs, and covering burn wounds with PF-127 have all been given sufficient importance worldwide [13].

Renowned in traditional Chinese medicine, *Polygonum cuspidatum* (PC) Siebold et Zuccarinii, has long been used in the treatment of inflammatory disorders, hepatitis, malignancies, and diarrhea in Asian nations, including China, Japan, and Korea [14]. PC extracts from different parts have vasorelaxant, antithrombotic, anti-angiogenesis, and anti-photoaging properties. Specifically, the PC plant root has been employed as an emmenagogue, sedative, and in folk medicine for diuresis. Additionally, the ethanol extract of PC root and rhizome has demonstrated excellent anti-inflammatory effects, with emodin as an active component in the extract responsible for these activities [15]. Pharmacological effects have proven that this effect of PC extract might be related to the increased expression of mucin-4 and the inhibition of oxidative stress and inflammation [16]. PC and its extracts also regulate carbohydrate and lipid metabolism. α-glucosidase and protein-tyrosine phosphatase 1B (PTP1B) are essential in insulin metabolism [17]. However, PC extract has not been incorporated into SnO_2_ NP biosynthesis to date. The aim of the current research is the green synthesis of Pluronic-F-127-encapsulated SnO_2_ NPs by utilizing the root extract of the *Polygonum cuspidatum* plant. In addition, anticancer activity against HepG2 cell lines is evaluated.

## 2. Results and Discussion

### 2.1. Characterization of Green Pluronic F-127 SnO_2_ NPs

#### 2.1.1. UV–Visible Spectrometry Analysis

The UV–vis absorption spectra of the Pluronic F-127 SnO_2_ NPs are depicted in Figure 1a. Strong absorption peaks were noticed at around 281 and 399 nm, which appear due to the band gap transitions and ultimately can lead to the crystalline structure of the formed Pluronic F-127 and SnO_2_ NPs [18].

#### 2.1.2. FTIR Spectrum Analysis

The surface chemistry of the NPs was examined by FTIR spectroscopy to determine the functional groups that may be in charge of their reduction, capping, and stability [19]. Figure 1b displays the FTIR spectrum of green Pluronic F-127 SnO_2_ NPs. Broad absorption in the frequency range 3419 cm^−1^ was observed in the NP sample, which is attributable to the O-H stretching of the hydroxyl groups [20]. Furthermore, functional groups at 2924 cm^−1^, 2854 cm^−1^, 1647 cm^−1^, and 1384 cm^−1^ can be attributed to asymmetric, symmetric C-H stretching, C-O stretching, and O-H bending, respectively. The aromatic C-C in-ring stretching and NH_2_ deformation stretching are related to the detected peaks at 1362 cm^−1^. The O-Sn-O stretching vibration is associated with the metal-oxide (M-O) stretching at 495 cm^−1^ [21]. The PF-127 molecules present on the SnO_2_ surface matrices led to the development of well-encapsulated NPs, which was supported by the FTIR data. The electrostatic interaction between the Pluronic F-127 and SnO_2_ NPs resulted in this encapsulation.

#### 2.1.3. Photoluminescence Spectroscopy Examination

Photoluminescence (PL) spectroscopy of the synthesized green SnO_2_ NPs was performed since peaks in the PL spectrum are typically linked to band-to-defect-level or band-to-band transitions or the recombination of excitons [22]. Figure 1c depicts the photoluminescence spectrum of synthesized green SnO_2_ NPs with an excitation wavelength of 325 nm. SnO_2_ NPs have PL emission values at 368, 396, 414, 436, 479, and 523 nm. The transition from the shallow donor level created by the O_2_ vacancies V_O_° is what caused the UV emission at 368 and 396 nm [23]. Due to the proximity of V_O_^++^ to VB, an electron leaves the conduction band and enters the acceptor level, producing the violet emission at 400 and 414 nm [24,25]. The shift from the donor level generated by VO^+^ to the valence band and the electron shift from O_2_ vacancies V_O_° to doubly ionized O_2_ vacancies V_O_^++^ are both responsible for the blue emission that is noted at 436 nm, and 479 nm. Oxygen vacancies and O_Sn_ defects are attributed for the green emission noticed at 523 nm. 

#### 2.1.4. Morphology and Chemical Composition

The surface morphology of the SnO_2_ NPs was identified using FESEM/TEM/SAED patterns, as demonstrated in Figure 2 and Figure 3. The morphological characteristics of the nanoparticles have been extensively studied using microscopy-based methods, including SEM and TEM. These methods can also be employed to determine the average size of the obtained NPs [25,26]. The formation of SnO_2_ NPs with a slightly agglomerated spherical shape was discovered using FESEM and TEM examination. The findings are in good conformity with the XRD-determined average particle size of 40–50 nm. The selected area of the electron diffraction pattern was utilized to confirm the formation of the SnO_2_ tetragonal rutile crystalline phase (Figure 2e) [27]. The chemical content of NPs was determined using an EDAX spectrum, as can be seen in Figure 3c. The SnO_2_ NPs were discovered to have the following atomic percentages: 12.73% (C), 26.47% (Sn), and 60.81% (O), respectively.

#### 2.1.5. XRD Study

Figure 4a depicts the XRD pattern of the Pluronic-F-127-encapsulated SnO_2_ NPs. The XRD peaks were noticed at angles (2θ) of 26.44°, 33.74°, 37.78°, 38.83°, 51.64°, 54.65°, 57.77°, 61.80°, 64.70°, 65.91°, 71.06°, and 78.61° and correspond to the (110), (101), (200), (111), (211), (220), (002), (310), (112), (301), (202), and (321) hkl planes of the SnO_2_ NPs, respectively, which matched with the tetragonal rutile-type SnO_2_ (space group P42/mnm) crystallite structure (JCPDS No. 41-1445) [27].
$$\mathrm{Average}\;\mathrm{crystallite}\;\mathrm{size}\;(D)=\frac{0.9\lambda }{\beta \cos\theta }$$
where ‘λ’- X-ray wavelength (1.54060 Å), ‘θ’- Bragg’s diffraction angle, and ‘β’- angular peak width at half maximum (radians). SnO_2_ NPs have an average crystallite size of 45 nm. Because the NPs were encapsulated in water, the DLS particle size was greater than the XRD findings. According to the DLS spectrum (Figure 4b), the hydrodynamic diameter of the NPs was 165 nm.

### 2.2. Assessment of Anticancer Activity

#### 2.2.1. Cytotoxicity Analysis by MTT Assay

One of the deadliest illnesses in the world is cancer, which claims more lives in developing nations [28]. The most prevalent major liver cancer is hepatocellular carcinoma, which is also the fifth most frequently diagnosed cancer worldwide [29]. It has been proven that oxidative stress contributes significantly to the onset and course of numerous diseases that affect a variety of important organs, especially cancer [30]. Owing to a very constrained therapeutic index, poor solubility, and toxicity towards normal tissues, existing chemotherapeutic treatment is rather limited and insufficient [31]. In this view, polymer-based drug delivery systems have been proposed as a means of ensuring highly localized chemotherapeutic concentration of drugs in malignant areas with little impact on healthy cells [32]. 

The biosynthesized SnO_2_ NPs were assessed for their cytotoxicity against HepG2 and Chang liver cell lines (normal liver cell lines) at various concentrations (2, 4, 8, 16, 20, and 40 µg/mL). The NPs demonstrated a substantial cytotoxicity against HepG2 cells with an IC_50_ concentration of 18 µg/mL. Moreover, the cell viability was observed to decrease with increasing NP concentration (Figure 5). However, the NPs failed to exhibit any cytotoxicity against normal cell lines, which indicated their non-toxic nature towards healthy cells. The accumulation of oxidative stress from ROS could be attributed to the observed cytotoxic effect [33]. According to a number of investigations, green-synthesized SnO_2_ NPs with several plant-derived extracts have shown remarkable anticancer capabilities [34,35]. To enhance the nanoparticle stability in an aqueous system, Pluronic F-127 was utilized to encapsulate them [36]. Recent research has also shown that Pluronic F-127 can react with cancer cells that are resistant to many drugs, thereby chemo-sensitizing those cells [37].

#### 2.2.2. Apoptotic Cell Death Analysis

A mixture of ethidium bromide and acridine orange dyes was used for staining the HepG2 cells at two concentrations of green SnO_2_ NPs (30 and 40 µg/mL), and then fluorescence microscopy was used for analysis (Figure 6). Green fluorescence indicates the presence of live cells that are not apoptotic. Pluronic-F-127-coated SnO_2_-NP-treated cells displayed yellow and orange fluorescence, showing condensed nuclei and necrotic cells, respectively, indicating early and late apoptosis. The suggested apoptotic mechanism for green-NP-induced cell death is as follows: the green SnO_2_ NPs induces loss of membrane integrity and leakage of cells. Further, the Sn^4+^ ions are released at the cell membrane, which triggers ROS formation, leading to oxidative stress. The generated ROS cause DNA damage and mitochondrial dysfunction, which ultimately results in apoptotic cell death [38].

#### 2.2.3. Cell Cycle Analysis

The cell cycle arrest in the control and green-synthesized SnO_2_-NP-treated HepG2 cells was investigated using a flow cytometry study, and the findings are demonstrated in Figure 7. The treatment with the green-synthesized SnO_2_ NPs at concentrations of 30 and 40 µg demonstrated an increased proportion of cells at G0/G1 stage. Furthermore, the cell proportions in the G2/M stage were decreased in the HepG2 cells after they were treated with green-synthesized SnO_2_ NPs. This finding revealed that green-synthesized SnO_2_ NP treatment demonstrated effective cell cycle arrest in the G0/G1 stages in the HepG2 cells (Figure 7). 

Understanding the various stages of the cell cycle is essential in order to inhibit tumor cell proliferation. It was already described that the tumor cell cycle can be inhibited using exogenous substances, including NPs [39,40]. Several signaling proteins participate in cell cycle regulation, which regulates tumor cell growth [41]. The cell cycle checkpoints are generally activated in response to DNA injury and replication stress. In normal cell division, each checkpoint is controlled strictly, and cells continue to divide rapidly if any of those checkpoints are disrupted. The inhibition of the cell cycle is said to be a hopeful treatment option for tumor treatment [42]. The outcomes of the present work found that green-synthesized SnO_2_ NPs inhibited the cell cycle in HepG2 cells at the G0/G1 stages. An earlier report already highlighted the cell cycle inhibition in several tumor cells by the green-synthesized NPs [43,44], which is in line with the current results.

#### 2.2.4. Cell Apoptosis Analysis by Flow Cytometry

Apoptosis initiation in cancer cells is thought to be a vital approach for treating several types of cancer with anticancer medications [45,46]. Here, flow cytometry was employed to examine the proportions of apoptotic cells in both control and treated HepG2 cells, and the outcomes are displayed in Figure 8. The findings confirm the occurrence of increased percentages of apoptotic cells in the green-synthesized SnO_2_-NP-treated HepG2 cells when compared with the control. Several previous studies using green-synthesized NPs have demonstrated apoptosis induction in tumor cells [47,48], including HepG2 cells. These studies supported the findings of the present study.

#### 2.2.5. Analysis of Endogenous ROS Accumulation

ROS plays a key role by increasing oxidative stress and thereby facilitating apoptosis in tumor cells. Furthermore, a moderate ROS level is essential for the regulation of normal cellular mechanisms; however, excessive ROS production causes DNA and cell damage and triggers cell death [49]. Therefore, the promotion of endogenous ROS accumulation in cancer cells is a key technique for developing antitumor agents and encourages oxidative stress-mediated apoptosis [50]. 

The level of endogenous ROS accumulation in the control and exposed HepG2 cells was examined, and the findings are shown in Figure 9. The control cells revealed poorly fluoresced cells; contrastingly, the green-synthesized SnO_2_-NP-treated HepG2 cells demonstrated higher fluorescence, which unveils higher endogenous ROS accumulation in the HepG2 cells. The induction of oxidative stress-mediated apoptosis in tumor cells, including HepG2 cells, was already reported in several earlier studies [51,52]. A recent report from Safwat et al. [53] has shown that green-synthesized NPs encourage ROS-mediated apoptosis G0/G1 cell cycle arrest in HepG2 cells, which supports the findings of the current work.

#### 2.2.6. Analysis of Apoptotic Protein Expression Levels

As seen in Figure 10, the HepG2 cells treated with green-synthesized SnO_2_ NPs demonstrated increased pro-apoptotic protein expressions such as cyt-c, p53, Bax, and caspase-3, -8, and -9 when compared with the control. Furthermore, treatment with green-synthesized SnO_2_ NPs reduced Bcl-2 expression in HepG2 cells. These findings revealed that the treatment with green-synthesized SnO_2_ NPs increases apoptosis in HepG2 cells via increased pro-apoptotic protein expression. The defects in apoptosis are a key contributor to continued cell division and tumor progression. If the neoplastic cells are apoptotic-resistant due to genetic causes, they will undergo oncogenic transformation. During cancer progression, the defects in apoptosis encourage the prolonged life span of cancer cells, survival under stress conditions, and tumor metastasis, which further result in apoptosis resistance in tumor cells [54].

Tumor cell fate and response to therapies are governed by the expression of pro- and anti-apoptotic genes. The increased Bcl-2 expression and reduced Bax expression are responsible for the emergence of apoptotic resistance to treatments [55]. Several chemotherapeutic drugs hinder the viability of tumor cells by blocking Bcl-2 expression and promoting Bax, p53, cyt-c, ad caspase-3, -8, and -9 expressions in cancer cells [56]. Caspase enzymes are key regulators of apoptosis [57]. The outcomes of this work revealed that the green-synthesized SnO_2_ NPs effectively promoted the expressions of Bax, Cyt-C, p53, and caspase-3, -8, and -9 in the HepG2 cells while decreasing Bcl-2. Hence, it was clear that green-synthesized SnO_2_ NPs can trigger apoptosis in HepG2 cells by encouraging pro-apoptotic protein expressions. The findings of the current work were supported by the earlier report by Hanna and Saad [58].

#### 2.2.7. Analysis of PI3K/Akt/mTOR Pathway

The influence of green-synthesized SnO_2_ NPs on the inhibition of PI3K/Akt/mTOR axis in the HepG2 cells were investigated by RT-PCR, and the findings were revealed in Figure 11. After the treatment with green-synthesized SnO_2_ NPs, the HepG2 cells demonstrated decreased expressions of Akt, mTOR, and PI3K, which confirms the inhibition of the PI3K/Akt/mTOR axis in HepG2 cells. The aberrant activation of PI3K/Akt/mTOR signaling is often said to be linked with tumorigenesis and also to facilitate tumor cell resistance to treatments. This signaling axis tightly regulates several pro-tumorigenic effects, such as cell growth, survival, invasion, and metastasis [59]. The activated PI3K/Akt/mTOR pathway has been highlighted for developing oncogenic phenotypes [60]. Therefore, the inhibitors of the PI3K/Akt/mTOR axis are currently being explored for the successful treatment of cancers [61]. Surprisingly, our findings showed that green-synthesized SnO_2_ NPs effectively inhibited PI3K/Akt/mTOR signaling in HepG2 cells (Figure 12).

One of the future perspectives of Pluronic-F-127-encapsulated SnO_2_ nanoparticles is their use in targeted drug delivery. These nanoparticles can be functionalized with ligands that specifically bind to certain cells, tissues, or organs, allowing for precise drug delivery and reduced side effects. Furthermore, the controlled release of drugs from these nanoparticles can be triggered by external stimuli such as pH, temperature, or light, which makes them particularly useful for cancer treatment.

## 3. Materials and Methods

### 3.1. Materials

Tin (II) chloride (SnCl_2_), Pluronic F-127, and other were obtained from Sigma-Aldrich, St. Louis, MO, USA. The culture consumables and media were attained from Sigma Aldrich, St. Louis, MO, USA. The ELISA kits were provided by Abcam, Cambridge, UK.

### 3.2. Preparation of Polygonum Cuspidatum Root Extract

For extraction, 10 g of fresh *Polygonum cuspidatum* root was combined with ethanol (100 mL) and boiled at 80 °C for 20 min. A filter paper was employed to filter the obtained extract, and the liquid was recovered in an Erlenmeyer flask and kept at 37 °C for subsequent use.

### 3.3. Preparation of Pluronic F-127 Encapsulated SnO_2_ NPs

For preparing PF-127-encapsulated SnO_2_ NPs, 0.1M of Tin (II) chloride (SnCl_2_) and 0.5 g of PF-127 was dissolved in 100 mL of *Polygonum cuspidatum root* extract. The resultant green-yellow-colour homogeneous mixture was constantly stirred at 80 °C for 5 h. The formed white depositions was let to dehydrate at 120 °C for 1 h. The developed PF-127-coated SnO_2_ NPs in powder form was calcined at 800 °C for 5 h and obtained for subsequent assays.

### 3.4. Characterization of Pluronic F-127 coated SnO_2_ NPs

The PF-127-coated SnO_2_ NPs was characterized with the help of an XRD (Bruker-AXS D5005). The study was done using Cu-Kα radiation at λ = 0.1541 nm wavelength, and scanned at 2θ between 20 and 90° angle.

A DLS examination was conducted to estimate the particle size of the formulated NPs. The study was done using formulated NPs in 90° scattering angle at 25 °C.

The morphology and the elemental profile of the PF-127-coated SnO_2_ NPs were observed with the help of an FE-SEM study utilising a Hitachi s-4800II build with EDAX. At an applied potential of 20 KV, an images were taken.

To further examine the morphology of the NPs, TEM (Jeol Jem-2010F) was employed. Briefly, the formulated NPs was spread on the copper grid, and then electronic radiation was utilized to light it in vacuum. In addition, a beam of electrons was utilized to take the images.

The FTIR (NicoletiS50) study was utilized to study the functional groups occur in the formulated PF-127 encapsulated SnO_2_ NPs. The spectra was studied using the reflectance technique. The formulated NPs was powdered with KBr at 1:100 ratio and spread into a discs. Then discs were put right into the spectrometer, where they were scanned between 400 and 4000 cm^−1^.

The formulated NPs was studied by UV–vis spectroscope (Shimadzu UV-2550, Shimadzu, Kyoto, Japan) to detect the development of NPs. The fabricated NPs was investigated in the 1200–200 nm range.

The optical effects of the formulated NPs were evaluated by photoluminescence spectroscopy (Roithner Laser Technik). The spectrum was studied using 350–550 nm spectral area, which was acquired at λexc = 470 nm. The energy band gap was studied by Tauc plotting using below formula:(hνα)1/n = A (hν − Eg)
where h represents Planck’s constant, A represents the constant proportion, α represents the absorption coefficient, v represents the vibration frequency, Eg represents the band gap, and n represents the sample’s transition nature.

### 3.5. In Vitro Anticancer Effects of Pluronic-F-127-Coated SnO_2_ NPs

#### 3.5.1. Maintenance of Human Liver Carcinoma Cells-HepG2

HepG2 cells (human liver carcinoma cells) was received from the ATCC, USA, and kept in DMEM medium with 10% FBS, and 1% antimycotic cocktail.

#### 3.5.2. Cytotoxicity Assay

A cytotoxicity test was employed to assess the influence of PF-127-coated SnO_2_ NPs on the HepG2 cell growth. In a 24-wellplate, the HepG2 cells were grown for 24 h and then treated with several dosages of NPs (2, 4, 8, 16, 20, 40 µg/mL) 24 h. Then MTT reagent (5 mg/mL) was mixed to each well for 3 h at room temperature. Afterward, DMOS was utilized to dissolve the formed formazan depositions and finally absorbance at 540 nm was measured using microplate reader. The IC50 level of NPs was fixed using programme Originpro8 [62].

#### 3.5.3. Apoptosis Analysis

The dual staining method was done to examine the apoptosis inducing potential of Pluronic-F-127-coated SnO_2_ NPs against HepG2 cells. Cells were grown in 6-wellplate and treated with 30 and 40 μg/mL of Pluronic-F-127-coated SnO_2_ NPs for 24 h. Later, the cells was stained using 1:1 ratio of AO/EtBr mixture for 5 min. The cells was later rinsed with PBS and assessed using fluorescent microscope [63].

#### 3.5.4. Cell Cycle Analysis

To analyze the cell cycle, the treated HepG2 cells was gathered and treated with ethanol (70%) and incubated for 12 h. Then cells was rinsed with buffer and stained with 300 µL of staining solution (100 µL of PI and 0.08 mg/mL proteinase inhibitor). A total of 0.5 mg/mL RNase was mixed for 30 min. The developed fluorescence was detected using flow cytometry, and the proportion of nuclei in each phase (G1, S, and G2/M) were determined using MultiCycle software [64].

#### 3.5.5. Analysis of Apoptosis by Flow Cytometry

The incidence of apoptosis in control and treated cells were quantified using flow cytometry. A 6-well plate was used to grow the cells for 24 h, which were then treated with 20 and 30 µg of Pluronic-F-127-coated SnO_2_ NPs for 24 h. Later, cells was assessed using Annexin V-FITC/PI-Apoptosis Detection Kit to assess the proportion of apoptotic levels. The experiments were performed in triplicate using the manufacturer’s protocols (Abcam, Boston, MA, USA) [65].

#### 3.5.6. DCFH-DA Staining

The endogenous ROS accumulation level in the treated HepG2 cells was detected using DCFH-DA staining approach. The cells were cultured in a 24-wellplate and exposed to Pluronic-F-127-coated SnO_2_ NPs for 24 h. Later, DCFH-DA (10 μL) was added to each well for 1 h. Finally, the endogenous ROS accumulation level in the cells was detected using a fluorescent microscope [66].

#### 3.5.7. Assay of Apoptotic Biomarker Levels

The levels of apoptotic proteins such as cyt-c, p53, Bax, Bcl-2,and caspase-3, -8, and -9 in the cell lysates of the control and Pluronic-F-127-coated SnO_2_-NP-treated HepG2 cells were evaluated using the ELISA kits using the guidelines of the manufacturer (Thermo Fisher Scientific, Waltham, MA, USA).

#### 3.5.8. RT-PCR Analysis

By using RT-PCR, the PI3K/AKT/mTOR signaling molecule expressions in the control and Pluronic-F-127-coated SnO2-NP-treated HepG2 cells was determined. The RNA was separated from the cells using a TRIzol kit (Thermofisher, Waltham, MA, USA), and the purified RNA was applied to construct the cDNA. In the following step, gene expression was examined using a PCR kit as per the manufacturer’s guidelines (Takara, Gunma, Japan) and 5-fluorouracil was used as a positive control. The primers for AKT forward were as follows: 5′-GAAGGACGGGAGCAGGCGGC-3′; the primers for AKT reverse were as follows: 5′-CCTCCTCCAGGCAGCCCCTT-3′. The primers for mTOR forward were as follows: 5′-AGTGGACCAGTGGAAACAGG-3′; the primers for mTOR reverse were as follows: 5′-TTCAGCGATGTCTTGTGAGG-3′. The primers for PI3K forward were as follows: 5′-AACACAGAAGACCAATACTC-3′; the primers for PI3K reverse were as follows: 5′-TTCGCCATCTACCACTAC-3′. The gene expressions were standardized with the control gene GADPH by the 2−ΔΔCT method [67].

### 3.6. Statistical Analysis

The values are investigated using one-way ANOVA and Tukey’s post-test after each study was conducted in triplicate. A *p* < 0.05 was fixed as significant of the results, which are displayed as mean±SD of triplicates.

## 4. Conclusions

In summary, pluronic-coated SnO_2_ NPs using *Polygonum cuspidatum* root extract was formulated and characterized using several methods. Moreover, their anticancer properties was investigated. The crystallite size of green SnO_2_ NPs was 45 nm. A tetragonal rutile-type crystal structure of the NPs was observed. FESEM analysis confirmed the NPs’ spherical structure. The NPs revealed a substantial cytotoxicity against HepG2 cells but failed to exhibit cytotoxicity when applied to other liver cell lines. In HepG2 cells, NPs treatment triggered cell cycle arrest in the G0/G1 stage. Furthermore, dual staining and flow cytometry studies of NP-treated HepG2 cells revealed an increase in apoptosis incidences. The NP treatment increased oxidative stress in HepG2 cells and promoted pro-apoptotic proteins. NPs also downregulated the PI3K/Akt/mTOR signaling in HepG2 cells, which leads to apoptosis. Thus, green Pluronic-F-127-coated SnO_2_ NPs demonstrate enormous efficacy to be utilized as a talented anticancer agent. Additional studies will aim on analyzing the molecular mechanisms of in vivo and clinical trials in hepatocellular carcinoma (HepG2) cancer cells.

## Figures and Tables

**Figure 1 plants-12-01760-f001:**
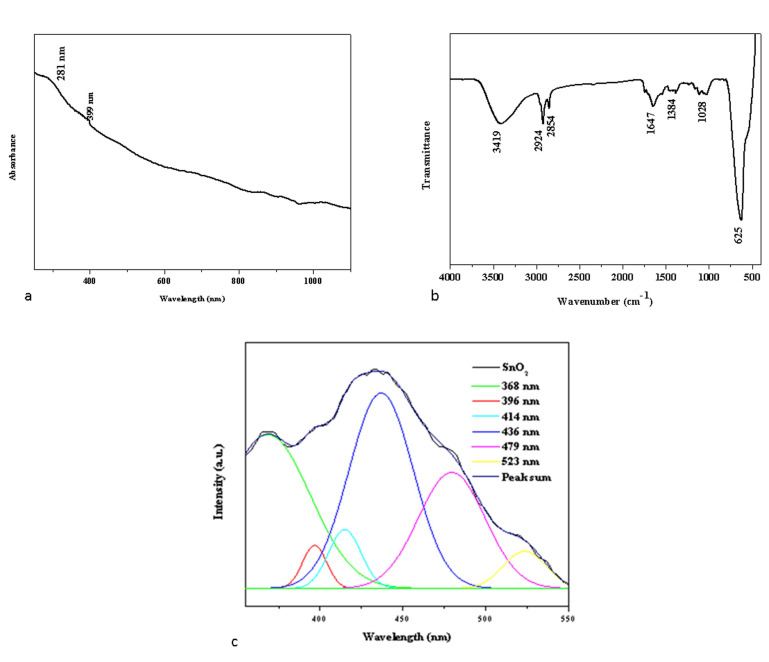
UV-vis spectrophotometry (**a**), FTIR transmittance vs. wavenumber chart, (**b**) and PL spectrum (**c**) study of formulated Pluronic-F-127-coated SnO_2_ NPs.

**Figure 2 plants-12-01760-f002:**
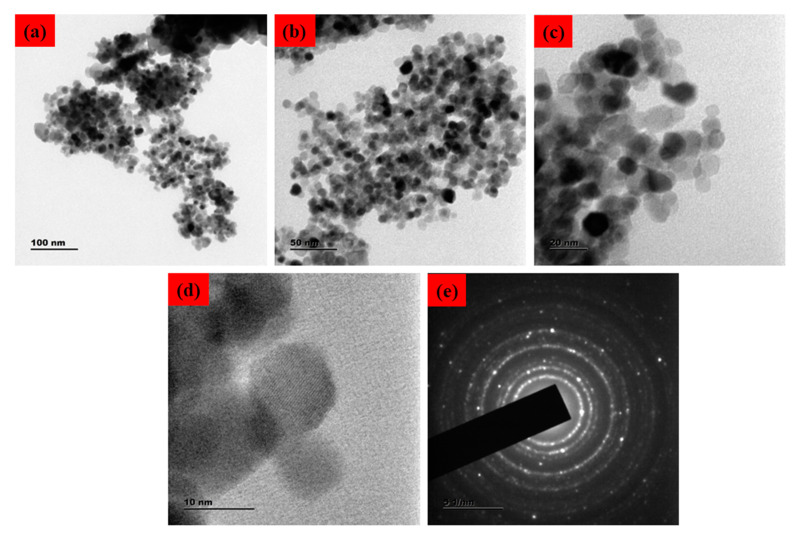
TEM micrographs of Pluronic-F-127-coated SnO_2_ NPs. Lower and higher magnification TEM photographs (**a**–**d**). SAED pattern of Pluronic-F-127-coated SnO_2_ NPs (**e**).

**Figure 3 plants-12-01760-f003:**
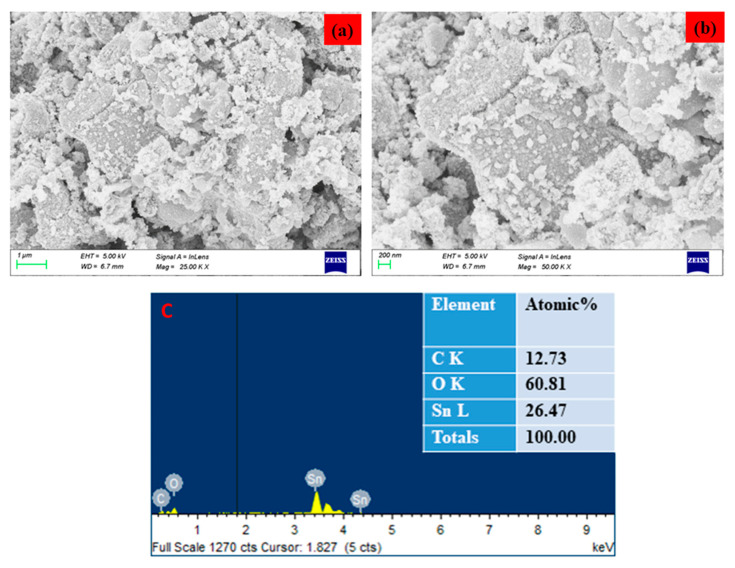
FESEM image of Pluronic-F-127-coated SnO_2_ NPs. Lower (**a**) and higher (**b**) magnification. Elements, weight %, and atomic % of the composition obtained by EDX (**c**).

**Figure 4 plants-12-01760-f004:**
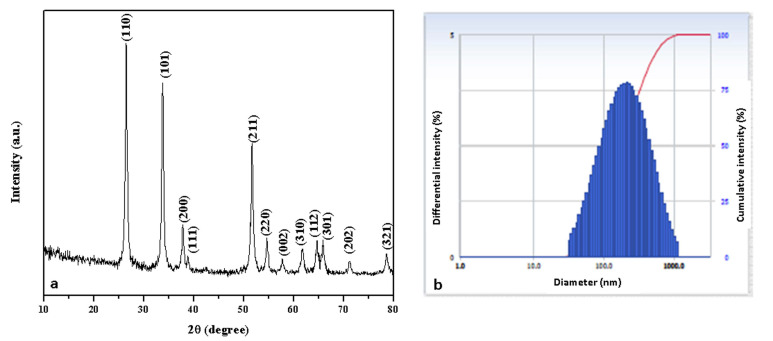
XRD pattern (**a**) and DLS pattern (**b**) of Pluronic-F-127-coated SnO_2_ NPs.

**Figure 5 plants-12-01760-f005:**
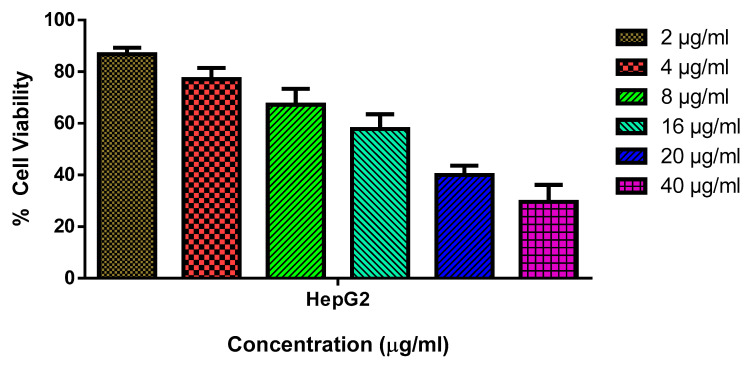
HepG2 cells are cytotoxic to SnO_2_ NPs coated with pluronic F-127. SnO_2_ NPs were used at different concentrations (2.0–40 µg/mL) for 24 h on HepG2 cell lines. An MTT study was employed on the cells, and their values are given as a mean ± SD of three separate assays.

**Figure 6 plants-12-01760-f006:**
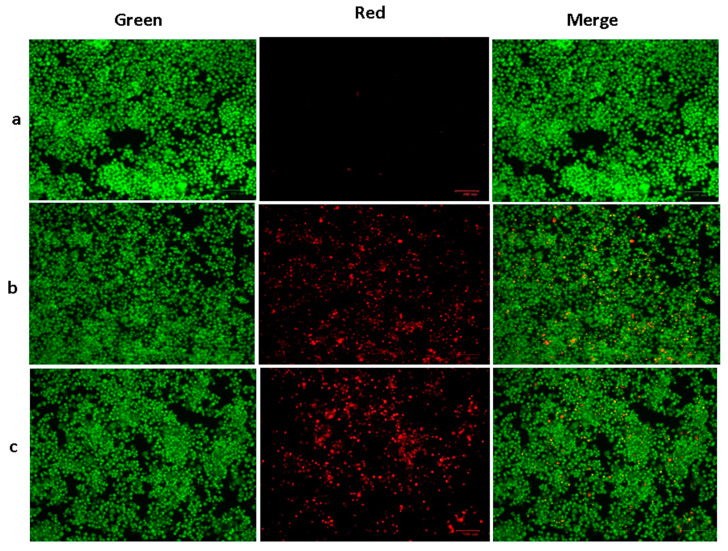
SnO_2_ nanoparticles coated with pluronic F-127 were shown to reduce cell apoptosis in liver cancer cells for 24 h after being applied. AO and EtBr (1:1) were stained in the cells, then investigated using fluorescent microscope. The control cells revealed green fluorescence, which denotes the live cells with absence of apoptosis. SnO_2_ NPs coated with Pluronic F-127 exhibited yellow/orange fluorescence, indicating early and late apoptosis, respectively. Control (**a**) (untreated cells), Pluronic-F-127-coated SnO_2_-NP-treated cells; 30 µg/mL dosage (**b**) and 40 µg/mL dosage (**c**).

**Figure 7 plants-12-01760-f007:**
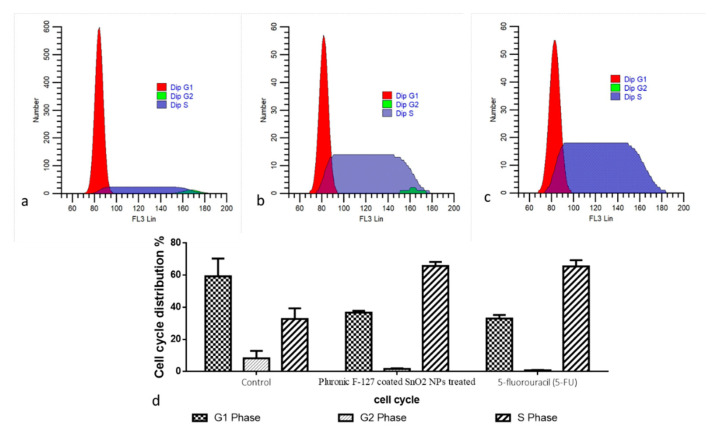
Analysis of cell cycle using flow cytometry. HepG2 cells were treated with IC50 concentration of Pluronic-F-127-coated SnO_2_ NPs (**b**) for 48 h and positive control 5-fluorouracil (5-FU) (**c**) with the concentration of 5µM/mL compared to the control (**a**). Cell cycle pattern and apoptosis distribution; percentage of cell cycle distribution (**d**).

**Figure 8 plants-12-01760-f008:**
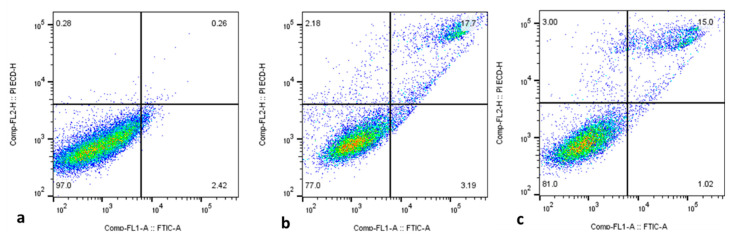
Flow cytometry analysis of HepG2 cancer cells treated for 48 h with the IC50 concentration of Pluronic-F-127-coated SnO_2_ NPs using Annexin-V/-FITC/PI. These data come from representative experiments conducted in at least two independent tests. The lower left quadrant (Annexin-V/PI) denotes the live cell proportions, while the lower right (Annexin-V+/PI) denotes the early apoptotic cell proportions, and upper (PI+) quadrants represent necrotic/secondary necrotic cells. Pluronic-F-127-coated SnO_2_-NP-treated cells (**a**), 30 µg/mL dosage (**b**) and 40 µg/mL dosage (**c**).

**Figure 9 plants-12-01760-f009:**
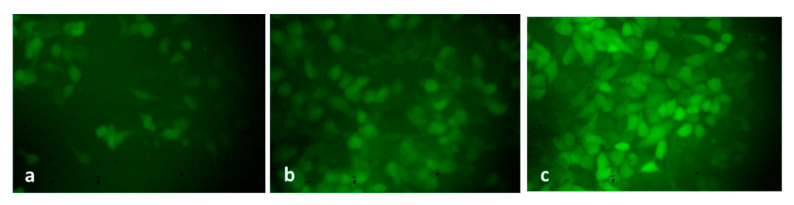
A fluorescent microscopic images of intracellular ROS generation induced by Pluronic-F-127-coated SnO_2_ NPs stained with DCF-DA. Control cells (**a**) cells treated with IC_25_ and (**b**) and IC50 (**c**) of Pluronic-F-127-coated SnO_2_ NPs. Magnification: 20X.

**Figure 10 plants-12-01760-f010:**
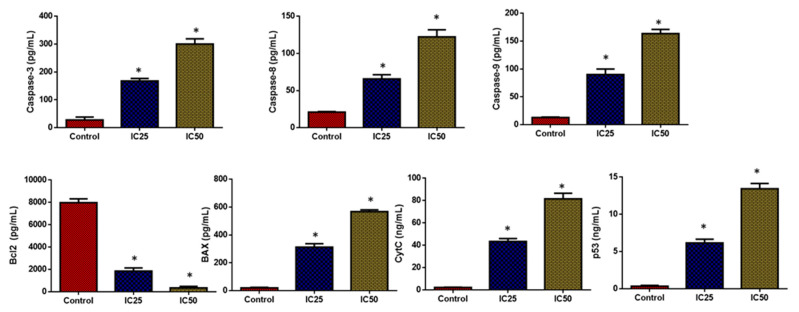
SnO_2_ NPs coated with pluronic F-127 inhibit proliferation and promote apoptosis in HepG2 cells. By using ELISA, we measured caspase-3, 8, 9, and Bax levels in the cells, as well as Bcl-2, CytC, and P53 levels. Each test was done in triplicates. The outcomes are presented as mean±SD of triplicates. ‘*’ *p* < 0.05.

**Figure 11 plants-12-01760-f011:**
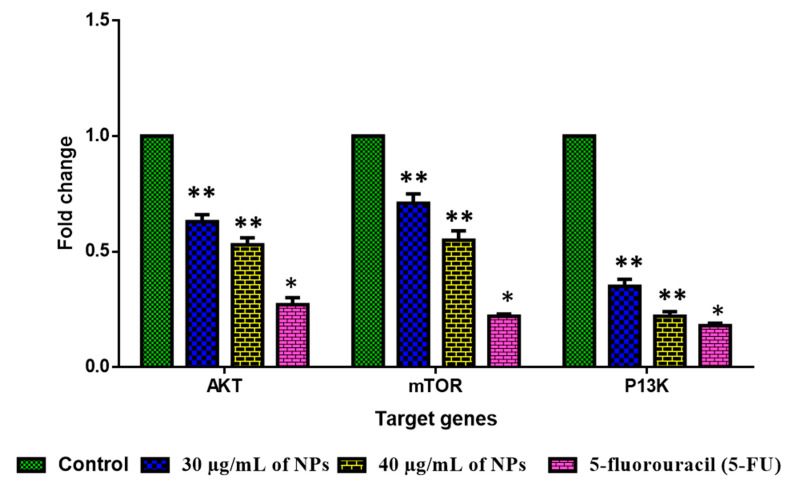
An analysis of PI3K/AKT/mTOR pathway in HepG2 cells with Pluronic-F-127-coated SnO_2_ nanoparticles. Using SPSS software V 20, one-way ANOVA and Tukey postdoc test was employed to analyze the values and represented as mean ± SD of triplicates. ‘**’ *p* < 0.05 and ‘*’ *p* < 0.01 compared with control.

**Figure 12 plants-12-01760-f012:**
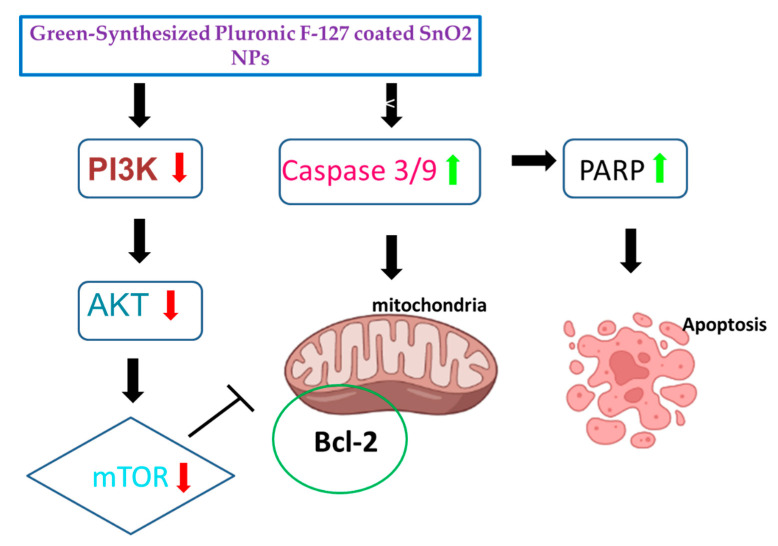
Schematic representation of proposed mechanism of Pluronic-F-127-coated SnO_2_ NPs on HepG2 cells.

## Data Availability

All the data associated with this work are available from the corresponding author based upon request.

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
