# Peer review of "Pluronic-F-127-Passivated SnO2 Nanoparticles Derived by Using Polygonum cuspidatum Root Extract: Synthesis, Characterization, and Anticancer Properties"

_plants, 2023, doi:10.3390/plants12091760_

Round 1
Reviewer 1 Report
- It is an interesting work with vague presentation
- There are 12 affiliations but only 10 are described. Please do check.
- Abstract is vague. Need to include why tin oxide has been synthesized instead of other metal oxide nanoparticles. Mention novelty and significance of the study
- Introduction need to follow bottleneck approach. Explore previous studies of biosynthesized Tin oxide nanoparticles, compare and come up with novelty as well as significance of the study
- Update references to recent studies in the introduction
- Materials and methods section should be before results and discussion section.
- Match the XRD data with the JCPDS file data of Tin oxide and superimpose the image
- The names of a bacteria should be changed through out the manuscript. When it appears first, need to expand their names and their names should be in italics. Also, it should be E. coli and S. pneumonia (add a space after dot).
- In the discussion, explain mechanism of action of tin oxide. May be a proposed mechanism with a schematic will help.
- The word in-vitro should be in vitro
Author Response
RESPONSES TO THE EDITOR
We authors would like to express our sincere thanks to the editor and reviewer for considering our work for publication in "Plants". We have entirely revised the manuscript as advised by the editor and reviewers. The responses to the individual comments of the reviewers are explained clearly as follows.
1 Q: There are 12 affiliations but only 10 are described. Please do check.
R: The author thanks the reviewers for valuable suggestions to improve the manuscript. As per the reviewer’s suggestion, we made the affiliation and author information correct, and the changes were marked in red text in the revised manuscript.
2 Q: - Abstract is vague. Need to include why tin oxide has been synthesized instead of other metal oxide nanoparticles. Mention novelty and significance of the study
R: As per the reviewer’s suggestion, we included the novelty and significance of the study as well as the purpose for which tin oxide has been synthesized instead of other metal oxide nanoparticles.
3 Q: Introduction need to follow bottleneck approach. Explore previous studies of biosynthesized Tin oxide nanoparticles, compare and come up with novelty as well as significance of the study
R: One of the main advantages of using plant extracts for the synthesis of nanoparticles is that it is an eco-friendly and sustainable method. Traditional methods of synthesis often involve the use of toxic chemicals and solvents, which can have negative impacts on the environment and human health. In contrast, the use of plant extracts as reducing and stabilizing agents is a safe and sustainable alternative. Moreover, the use of Polygonum cuspidatum root extract as a reducing and stabilizing agent for the synthesis of SnO2 nanoparticles is a novel approach. The plant extract contains various phytochemicals, such as flavonoids, tannins, and phenolic acids, which can act as reducing and stabilizing agents for the synthesis of nanoparticles. The use of plant extracts for the synthesis of nanoparticles is a relatively new area of research, and the use of Polygonum cuspidatum root extract as a reducing and stabilizing agent for the synthesis of SnO2 nanoparticles is a significant contribution to this field. The synthesized SnO2 nanoparticles also have potential applications in various fields, such as electronics, sensing, and catalysis. For example, SnO2 nanoparticles can be used as gas sensors to detect harmful gases, such as carbon monoxide and nitrogen dioxide. They can also be used as photocatalysts for the degradation of organic pollutants in water and air. In summary, the biosynthesis of SnO2 nanoparticles using root extracts of Polygonum cuspidatum is a novel and significant study that provides a sustainable and eco-friendly method for the synthesis of nanoparticles.
4 Q: Update references to recent studies in the introduction
R: As suggested by the reviewer, necessary modifications were carried out.
5 Q: Materials and methods section should be before results and discussion section.
R: The author thanks the reviewers for their valuable suggestions to improve the manuscript. The materials and methods section will come after the results and discussion section according to the journal format.
6 Q: Match the XRD data with the JCPDS file data of Tin oxide and superimpose the image
R: As suggested by the reviewer, the legends of the tables and figures are revised and highlighted in yellow.
7 Q: The names of bacteria should be changed throughout the manuscript. When it appears first, need to expand their names and their names should be in italics. Also, it should be E. coli and S. pneumonia (add a space after dot).
R: As suggested by the reviewer, necessary modifications to their names and italics have been made to the revised manuscript.
8 Q: - In the discussion, explain mechanism of action of tin oxide. May be a proposed mechanism with a schematic will help.
R: As suggested by the reviewer, we included the schematic representation of the mechanism of action of tin oxide in the discussion.
9 Q: - The word in-vitro should be in vitro
R: The author thanks the reviewers for their valuable suggestions to improve the manuscript. As suggested by the reviewer, the changes are highlighted in red text in the revised manuscript.
Reviewer 2 Report
Abozer et al., have studied about “Green synthesis, characterization, antibacterial and anticancer (HepG2) activity of plant-derived Pluronic F-127 encapsulated SnO2 nanoparticles using Polygonum cuspidatum root extract”.
The research work was well planned with adequate experiments even though author should address below mentioned corrections before accepted for publication in the journal.
Title of the manuscript has to be modified as well as abstract section based on the findings.
Lack of information about the pharmacological potential of Polygonum cuspidatum root extract.
Authors must go through the methods section carefully and do the necessary corrections in the appropriate places (cite the relevant references).
The manuscript mainly focuses on the anti-cancer potential of Pluronic F-127 encapsulated SnO2 nanoparticles with extract so I would suggest authors to remove the section of the results 2.2. Evaluation of antibacterial activity and 2.3. Evaluation of antioxidant activity from the manuscript. Those anti-bacterial ad antioxidant effects may mislead the conclusion of the manuscript therefore need to revise the manuscript accordingly.
Conclusion section should include the future prospective of research direction.
Author Response
RESPONSES TO THE EDITOR
We authors would like to express our sincere thanks to the editor and reviewer for considering our work for publication in "Plants". We have entirely revised the manuscript as advised by the editor and reviewers. The responses to the individual comments of the reviewers are explained clearly as follows.
1: The title of the manuscript has to be modified as well as the abstract section based on the findings.
R: The author thanks the reviewers for their valuable suggestions to improve the manuscript. We modified the abstract section in the revised manuscript based on the findings.
2 Q: - Lack of information about the pharmacological potential of Polygonum cuspidatum root extract
R: As suggested by the reviewer, we included the highlight section is revised. Resveratrol has been found to have antioxidant, anti-inflammatory, and anti-cancer properties. It has also been studied for its potential to prevent or treat a range of health conditions, including cardiovascular disease, diabetes, obesity, and neurodegenerative diseases. In addition to resveratrol, Polygonum cuspidatum root extract contains other compounds that may also have pharmacological potential. For example, emodin, a natural anthraquinone found in the root, has been shown to have anti-inflammatory and anti-tumor properties. Another compound, polydatin, has been studied for its potential to protect against liver damage and improve cognitive function. Overall, there is growing interested in the pharmacological potential of Polygonum cuspidatum root extract, particularly for its potential in preventing or treating chronic diseases. However, more research is needed to fully understand its mechanisms of action and potential therapeutic applications.
3 Q: Authors must go through the methods section carefully and make the necessary corrections in the appropriate places (cite the relevant references).
R: As suggested by the reviewer, we rephrased the methods section carefully and do the necessary corrections in the appropriate places with relevant citations.
4 Q: The manuscript mainly focuses on the anti-cancer potential of Pluronic F-127 encapsulated SnO2 nanoparticles with extract so I would suggest authors remove the section of the results 2.2. Evaluation of antibacterial activity and 2.3. Evaluation of antioxidant activity from the manuscript. Those anti-bacterial ad antioxidant effects may mislead the conclusion of the manuscript therefore need to revise the manuscript accordingly.
R: As per the reviewer’s suggestion, we removed both sections of the results 2.2. Evaluation of antibacterial activity and 2.3. Evaluation of antioxidant activity from the manuscript. We improved the discussion and conclusion in the revised manuscript.
5 Q: The conclusion section should include the future perspective of the research direction
R: As per the reviewer’s suggestion, we included the Future perspective of Pluronic F-127 encapsulated SnO2 nanoparticles in the discussion and conclusion section. The sentences were red.
Round 2
Reviewer 1 Report
The authors have revised the manuscript according to the reviewer comments. There are few grammar errors and typos, which should be avoided before publication.